# Learning Python Code Suggestion with a Sparse Pointer Network

**Avishkar Bhoopchand, Tim Rocktäschel, Earl Barr & Sebastian Riedel**
Department of Computer Science
University College London
`avishkar.bhoopchand.15@ucl.ac.uk,`
`{t.rocktaschel,e.barr,s.riedel}@cs.ucl.ac.uk`

## Abstract

To enhance developer productivity, all modern integrated development environments (IDEs) include *code suggestion* functionality that proposes likely next tokens at the cursor. While current IDEs work well for statically-typed languages, their reliance on type annotations means that they do not provide the same level of support for dynamic programming languages as for statically-typed languages. Moreover, suggestion engines in modern IDEs do not propose expressions or multi-statement idiomatic code. Recent work has shown that language models can improve code suggestion systems by learning from software repositories. This paper introduces a neural language model with a sparse pointer network aimed at capturing very long-range dependencies. We release a large-scale code suggestion corpus of 41M lines of Python code crawled from GitHub. On this corpus, we found standard neural language models to perform well at suggesting local phenomena, but struggle to refer to identifiers that are introduced many tokens in the past. By augmenting a neural language model with a pointer network specialized in referring to predefined classes of identifiers, we obtain a much lower perplexity and a 5 percentage points increase in accuracy for code suggestion compared to an LSTM baseline. In fact, this increase in code suggestion accuracy is due to a 13 times more accurate prediction of identifiers. Furthermore, a qualitative analysis shows this model indeed captures interesting long-range dependencies, like referring to a class member defined over 60 tokens in the past.

## 1 Introduction

Integrated development environments (IDEs) are essential tools for programmers. Especially when a developer is new to a codebase, one of their most useful features is code suggestion: given a piece of code as context, suggest a likely sequence of next tokens. Typically, the IDE suggests an identifier or a function call, including API calls. While extensive support exists for statically-typed languages such as Java, code suggestion for dynamic languages like Python is harder and less well supported because of the lack of type annotations. Moreover, suggestion engines in modern IDEs do not propose expressions or multi-statement idiomatic code.

Recently, methods from statistical natural language processing (NLP) have been used to train code suggestion systems from code usage in large code repositories (Hindle et al., 2012; Allamanis & Sutton, 2013; Tu et al., 2014). To this end, usually an $n$-gram language model is trained to score possible completions. Neural language models for code suggestion (White et al., 2015; Das & Shah, 2015) have extended this line of work to capture more long-range dependencies. Yet, these standard neural language models are limited by the so-called hidden state bottleneck, *i.e.*, all context information has to be stored in a fixed-dimensional internal vector representation. This limitation restricts such models to local phenomena and does not capture very long-range semantic relationships like suggesting calling a function that has been defined many tokens before.

To address these issues, we create a large corpus of 41M lines of Python code by using a heuristic for crawling high-quality code repositories from GitHub. We investigate, for the first time, the use of attention (Bahdanau et al., 2014) for code suggestion and find that, despite a substantial improvement

in accuracy, it still makes avoidable mistakes. Hence, we introduce a model that leverages long-range Python dependencies by selectively attending over the introduction of identifiers as determined by examining the Abstract Syntax Tree. The model is a form of pointer network (Vinyals et al., 2015a), and learns to dynamically choose between syntax-aware pointing for modeling long-range dependencies and free form generation to deal with local phenomena, based on the current context.

Our contributions are threefold: (i) We release a code suggestion corpus of 41M lines of Python code crawled from GitHub, (ii) We introduce a sparse attention mechanism that captures very long-range dependencies for code suggestion of this dynamic programming language efficiently, and (iii) We provide a qualitative analysis demonstrating that this model is indeed able to learn such long-range dependencies.

## 2 METHODS

We first revisit neural language models, before briefly describing how to extend such a language model with an attention mechanism. Then we introduce a sparse attention mechanism for a pointer network that can exploit the Python abstract syntax tree of the current context for code suggestion.

### 2.1 NEURAL LANGUAGE MODEL

Code suggestion can be approached by a language model that measures the probability of observing a sequence of tokens in a Python program. For example, for the sequence $S = a_1, \ldots, a_N$, the joint probability of $S$ factorizes according to

$$P_\theta(S) = P_\theta(a_1) \cdot \prod_{t=2}^{N} P_\theta(a_t \mid a_{t-1}, \ldots, a_1)$$ (1)

where the parameters $\theta$ are estimated from a training corpus. Given a sequence of Python tokens, we seek to predict the next $M$ tokens $a_{t+1}, \ldots, a_{t+M}$ that maximize Equation 1

$$\underset{a_{t+1}, \ldots, a_{t+M}}{\arg\max} \; P_\theta(a_1, \ldots, a_t, a_{t+1}, \ldots, a_{t+M}).$$ (2)

In this work, we build upon neural language models using Recurrent Neural Networks (RNNs) and Long Short-Term Memory (LSTM, Hochreiter & Schmidhuber, 1997). This neural language model estimates the probabilities in Equation 1 using the output vector of an LSTM at time step $t$ (denoted $\boldsymbol{h}_t$ here) according to

$$P_\theta(a_t = \tau \mid a_{t-1}, \ldots, a_1) = \frac{\exp\left(\boldsymbol{v}_\tau^T \boldsymbol{h}_t + b_\tau\right)}{\sum_{\tau'} \exp\left(\boldsymbol{v}_{\tau'}^T \boldsymbol{h}_t + b_{\tau'}\right)}$$ (3)

where $\boldsymbol{v}_\tau$ is a parameter vector associated with token $\tau$ in the vocabulary.

Neural language models can, in theory, capture long-term dependencies in token sequences through their internal memory. However, as this internal memory has fixed dimension and can be updated at every time step, such models often only capture local phenomena. In contrast, we are interested in very long-range dependencies like referring to a function identifier introduced many tokens in the past. For example, a function identifier may be introduced at the top of a file and only used near the bottom. In the following, we investigate various external memory architectures for neural code suggestion.

### 2.2 ATTENTION

A straight-forward approach to capturing long-range dependencies is to use a neural attention mechanism (Bahdanau et al., 2014) on the previous $K$ output vectors of the language model. Attention mechanisms have been successfully applied to sequence-to-sequence tasks such as machine translation (Bahdanau et al., 2014), question-answering (Hermann et al., 2015), syntactic parsing (Vinyals et al., 2015b), as well as dual-sequence modeling like recognizing textual entailment (Rocktäschel et al., 2016). The idea is to overcome the hidden-state bottleneck by allowing referral back to previous output vectors. Recently, these mechanisms were applied to language modelling by Cheng et al. (2016) and Tran et al. (2016).

Formally, an attention mechanism with a fixed memory $\boldsymbol{M}_t \in \mathbb{R}^{k \times K}$ of $K$ vectors $\boldsymbol{m}_i \in \mathbb{R}^k$ for $i \in [1, K]$, produces an attention distribution $\boldsymbol{\alpha}_t \in \mathbb{R}^K$ and context vector $\boldsymbol{c}_t \in \mathbb{R}^k$ at each time step $t$ according to Equations 4 to 7. Furthermore, $\boldsymbol{W}^M, \boldsymbol{W}^h \in \mathbb{R}^{k \times k}$ and $\boldsymbol{w} \in \mathbb{R}^k$ are trainable parameters. Finally, note that $\mathbf{1}_K$ represents a $K$-dimensional vector of ones.

$$\boldsymbol{M}_t = [\boldsymbol{m}_1 \ \ldots \ \boldsymbol{m}_K] \qquad\qquad \in \mathbb{R}^{k \times K} \tag{4}$$

$$\boldsymbol{G}_t = \tanh(\boldsymbol{W}^M \boldsymbol{M}_t + \mathbf{1}_K^T (\boldsymbol{W}^h \boldsymbol{h}_t)) \qquad\qquad \in \mathbb{R}^{k \times K} \tag{5}$$

$$\boldsymbol{\alpha}_t = \mathrm{softmax}(\boldsymbol{w}^T \boldsymbol{G}_t) \qquad\qquad \in \mathbb{R}^{1 \times K} \tag{6}$$

$$\boldsymbol{c}_t = \boldsymbol{M}_t \boldsymbol{\alpha}_t^T \qquad\qquad \in \mathbb{R}^k \tag{7}$$

For language modeling, we populate $\boldsymbol{M}_t$ with a fixed window of the previous $K$ LSTM output vectors. To obtain a distribution over the next token we combine the context vector $\boldsymbol{c}_t$ of the attention mechanism with the output vector $\boldsymbol{h}_t$ of the LSTM using a trainable projection matrix $\boldsymbol{W}^A \in \mathbb{R}^{k \times 2k}$. The resulting final output vector $\boldsymbol{n}_t \in \mathbb{R}^k$ encodes the next-word distribution and is projected to the size of the vocabulary $|V|$. Subsequently, we apply a softmax to arrive at a probability distribution $\mathbf{y}_t \in \mathbb{R}^{|V|}$ over the next token. This process is presented in Equation 9 where $\mathbf{W}^V \in \mathbb{R}^{|V| \times k}$ and $\boldsymbol{b}^V \in \mathbb{R}^{|V|}$ are trainable parameters.

$$\boldsymbol{n}_t = \tanh\left( \boldsymbol{W}^A \begin{bmatrix} \boldsymbol{h}_t \\ \boldsymbol{c}_t \end{bmatrix} \right) \qquad\qquad \in \mathbb{R}^k \tag{8}$$

$$\boldsymbol{y}_t = \mathrm{softmax}(\boldsymbol{W}^V \boldsymbol{n}_t + \boldsymbol{b}^V) \qquad\qquad \in \mathbb{R}^{|V|} \tag{9}$$

The problem of the attention mechanism above is that it quickly becomes computationally expensive for large $K$. Moreover, attending over many memories can make training hard as a lot of noise is introduced in early stages of optimization where the LSTM outputs (and thus the memory $\boldsymbol{M}_t$) are more or less random. To alleviate these problems we now turn to pointer networks and a simple heuristic for populating $\boldsymbol{M}_t$ that permits the efficient retrieval of identifiers in a large history of Python code.

## 2.3 SPARSE POINTER NETWORK

We develop an attention mechanism that provides a *filtered view* of a large history of Python tokens. At any given time step, the memory consists of context representations of the previous $K$ identifiers introduced in the history. This allows us to model long-range dependencies found in identifier usage. For instance, a class identifier may be declared hundreds of lines of code before it is used. Given a history of Python tokens, we obtain a next-word distribution from a weighed average of the sparse pointer network for identifier reference and a standard neural language model. The weighting of the two is determined by a controller.

Formally, at time-step $t$, the sparse pointer network operates on a memory $\boldsymbol{M}_t \in \mathbb{R}^{k \times K}$ of only the $K$ previous identifier representations (*e.g.* function identifiers, class identifiers and so on). In addition, we maintain a vector $\boldsymbol{m}_t = [\mathrm{id}_1, \ \ldots, \ \mathrm{id}_K] \in \mathbb{N}^K$ of symbol ids for these identifier representations (*i.e.* pointers into the large global vocabulary).

As before, we calculate a context vector $\boldsymbol{c}_t$ using the attention mechanism (Equation 7), but on a memory $\boldsymbol{M}_t$ only containing representations of identifiers that were declared in the history. Next, we obtain a pseudo-sparse distribution over the global vocabulary from

$$\boldsymbol{s}_t[i] = \begin{cases} \boldsymbol{\alpha}_t[j] & \text{if } \boldsymbol{m}_t[j] = i \\ -C & \text{otherwise} \end{cases} \tag{10}$$

$$\boldsymbol{i}_t = \mathrm{softmax}(\boldsymbol{s}_t) \qquad\qquad \in \mathbb{R}^{|V|} \tag{11}$$

where $-C$ is a large negative constant (*e.g.* $-1000$). In addition, we calculate a next-word distribution from a standard neural language model

$$\boldsymbol{y}_t = \mathrm{softmax}(\boldsymbol{W}^V \boldsymbol{h}_t + \boldsymbol{b}^V) \qquad\qquad \in \mathbb{R}^{|V|} \tag{12}$$

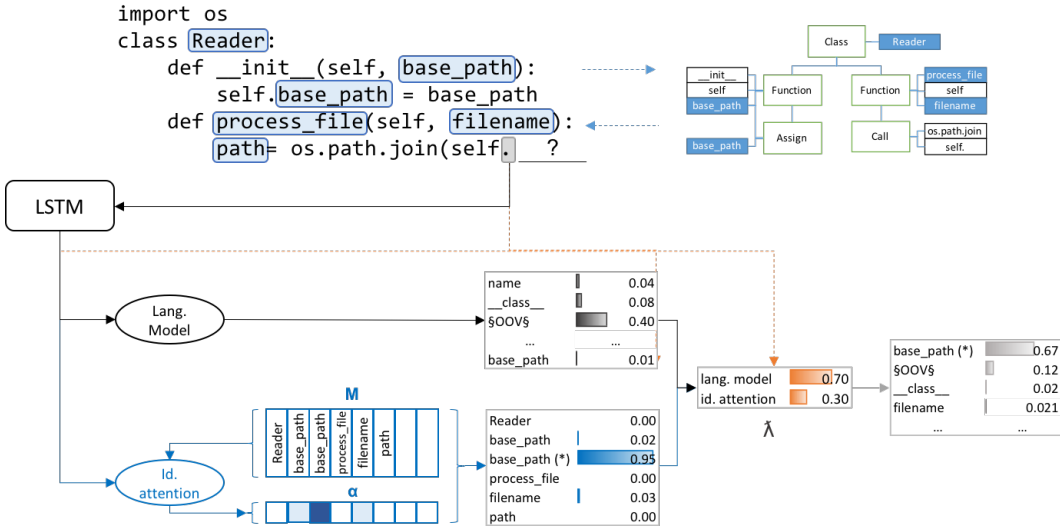

Figure 1: Sparse pointer network for code suggestion on a Python code snippet, showing the next-word distributions of the language model and identifier attention and their weighted combination through $\boldsymbol{\lambda}$

and we use a controller to calculate a distribution $\boldsymbol{\lambda}_t \in \mathbb{R}^2$ over the language model and pointer network for the final weighted next-word distribution $\boldsymbol{y}_t^*$ via

$$
\boldsymbol{h}_t^\lambda = \begin{bmatrix} \boldsymbol{h}_t \\ \boldsymbol{x}_t \\ \boldsymbol{c}_t \end{bmatrix} \qquad\qquad \in \mathbb{R}^{3k} \tag{13}
$$

$$
\boldsymbol{\lambda}_t = \mathrm{softmax}(\boldsymbol{W}^\lambda \mathbf{h}_t^\lambda + \boldsymbol{b}^\lambda) \qquad\qquad \in \mathbb{R}^2 \tag{14}
$$

$$
\boldsymbol{y}_t^* = [\boldsymbol{y}_t\ \boldsymbol{i}_t]\,\boldsymbol{\lambda}_t \qquad\qquad \in \mathbb{R}^{|V|} \tag{15}
$$

Here, $\boldsymbol{x}_t$ is the representation of the input token, and $\boldsymbol{W}^\lambda \in \mathbb{R}^{2\times 3k}$ and $\boldsymbol{b}^\lambda \in \mathbb{R}^2$ a trainable weight matrix and bias respectively. This controller is conditioned on the input, output and context representations. This means for deciding whether to refer to an identifier or generate from the global vocabulary, the controller has access to information from the encoded next-word distribution $\boldsymbol{h}_t$ of the standard neural language model, as well as the attention-weighted identifier representations $\boldsymbol{c}_t$ from the current history.

Figure 1 overviews this process. In it, the identifier base_path appears twice, once as an argument to a function and once as a member of a class (denoted by *). Each appearance has a different id in the vocabulary and obtains a different probability from the model. In the example, the model correctly chooses to refer to the member of the class instead of the out-of-scope function argument, although, from a user point-of-view, the suggestion would be the same in both cases.

## 3 LARGE-SCALE PYTHON CORPUS

Previous work on code suggestion either focused on statically-typed languages (particularly Java) or trained on very small corpora. Thus, we decided to collect a new large-scale corpus of the dynamic programming language Python. According to the programming language popularity website Pypl (Carbonnelle, 2016), Python is the second most popular language after Java. It is also the 3rd most common language in terms of number of repositories on the open-source code repository GitHub, after JavaScript and Java (Zapponi, 2016).

We collected a corpus of 41M lines of Python code from GitHub projects. Ideally, we would like this corpus to only contain high-quality Python code, as our language model learns to suggest code from how users write code. However, it is difficult to automatically assess what constitutes high-quality code. Thus, we resort to the heuristic that popular code projects tend to be of good quality, There are

Table 1: Python corpus statistics.

| Dataset | #Projects | #Files | #Lines | #Tokens | Vocabulary Size |
|---|---|---|---|---|---|
| Train | 489 | 118 298 | 26 868 583 | 88 935 698 | 2 323 819 |
| Dev | 179 | 26 466 | 5 804 826 | 18 147 341 | |
| Test | 281 | 43 062 | 8 398 100 | 30 178 356 | |
| Total | 949 | 187 826 | 41 071 509 | 137 261 395 | |

```python
from git import Repo

class PyRepo:
    def __init__(self, name, full_name, clone_url):
        self.name = name
        self.full_name = full_name
        self.clone_url = clone_url

    def checkout(self, output_directory):
        repo = Repo.clone_from(self.clone_url,
            os.path.join(output_directory, self.name))
        git = repo.git
        git.checkout(self.last_commit_sha)
```

```python
from git import Repo

class Class210:

    def __init__(self, arg1633, arg2343, arg233):
        self.attribute826 = arg1633
        self.attribute1352 = arg2343
        self.attribute172 = arg233

    def function1245(self, arg1316):
        var4311 = Repo.clone_from(self.attribute172,
            os.path.join(arg1316, self.attribute826))
        var142 = var4311.git
        var142.checkout(self.attribute471)
```

Figure 2: Example of the Python code normalization. Original file on the left and normalized version on the right.

two metrics on GitHub that we can use for this purpose, namely stars (similar to bookmarks) and forks (copies of a repository that allow users to freely experiment with changes without affecting the original repository). Similar to Allamanis & Sutton (2013) and Allamanis et al. (2014), we select Python projects with more than 100 stars, sort by the number of forks descending, and take the top 1000 projects. We then removed projects that did not compile with Python3, leaving us with 949 projects. We split the corpus on the project level into train, dev, and test. Table 1 presents the corpus statistics.

## 3.1 NORMALIZATION OF IDENTIFIERS

Unsurprisingly, the long tail of words in the vocabulary consists of rare identifiers. To improve generalization, we normalize identifiers before feeding the resulting token stream to our models. That is, we replace every identifier name with an anonymous identifier indicating the identifier group (class, variable, argument, attribute or function) concatenated with a random number that makes the identifier unique in its scope. Note that we only replace novel identifiers defined within a file. Identifier references to external APIs and libraries are left untouched. Consistent with previous corpus creation for code suggestion (*e.g.* Khanh Dam et al., 2016; White et al., 2015), we replace numerical constant tokens with $NUM$, remove comments, reformat the code, and replace tokens appearing less than five times with an $OOV$ (out of vocabulary) token.

## 4 EXPERIMENTS

Although previous work by White et al. (2015) already established that a simple neural language model outperforms an $n$-gram model for code suggestion, we include a number of $n$-gram baselines to confirm this observation. Specifically, we use $n$-gram models for $n \in \{3, 4, 5, 6\}$ with Modified Kneser-Ney smoothing (Kneser & Ney, 1995) from the Kyoto Language Modelling Toolkit (Neubig, 2012).

We train the sparse pointer network using mini-batch SGD with a batch size of 30 and truncated backpropagation through time (Werbos, 1990) with a history of 20 identifier representations. We use

Table 2: Perplexity (PP), Accuracy (Acc) and Accuarcy among top 5 predictions (Acc@5).

| Model | Train PP | Dev PP | Test PP | Acc [%] | | | Acc@5 [%] | | |
|---|---|---|---|---|---|---|---|---|---|
| | | | | All | IDs | Other | All | IDs | Other |
| 3-gram | 12.90 | 24.19 | 26.90 | 13.19 | – | – | 50.81 | – | – |
| 4-gram | 7.60 | 21.07 | 23.85 | 13.68 | – | – | 51.26 | – | – |
| 5-gram | 4.52 | 19.33 | 21.22 | 13.90 | – | – | 51.49 | – | – |
| 6-gram | 3.37 | 18.73 | 20.17 | 14.51 | – | – | 51.76 | – | – |
| LSTM | 9.29 | 13.08 | 14.01 | 57.91 | 2.1 | 62.8 | 76.30 | 4.5 | 82.6 |
| LSTM w/ Attention 20 | 7.30 | 11.07 | 11.74 | 61.30 | 21.4 | 64.8 | 79.32 | 29.9 | 83.7 |
| LSTM w/ Attention 50 | 7.09 | 9.83 | 10.05 | **63.21** | **30.2** | **65.3** | 81.69 | 41.3 | 84.1 |
| Sparse Pointer Network | 6.41 | **9.40** | **9.18** | 62.97 | 27.3 | 64.9 | **82.62** | **43.6** | **84.5** |

an initial learning rate of $0.7$ and decay it by $0.9$ after every epoch. As additional baselines, we test a neural language model with LSTM units with and without attention. For the attention language models, we experiment with a fixed-window attention memory of the previous $20$ and $50$ tokens respectively, and a batch size of $75$. We found during testing that the baseline models performed worse with the same batch size as the sparse pointer network of $30$. We therefore chose to report the stronger results obtained with a batch size of $75$.

All neural language models were developed in TensorFlow (Abadi et al., 2016) and trained using cross-entropy loss. While processing a Python source code file, the last recurrent state of the RNN is fed as the initial state of the subsequent sequence of the same file and reset between files. All models use an input and hidden size of $200$, an LSTM forget gate bias of $\mathbf{1}$ (Jozefowicz et al., 2015), gradient norm clipping of $5$ (Pascanu et al., 2013), and randomly initialized parameters in the interval $(-0.05, 0.05)$. As regularizer, we use a dropout of $0.1$ on the input representations. Furthermore, we use a sampled softmax (Jean et al., 2015) with a log-uniform sampling distribution and a sample size of $1000$.

## 5 RESULTS

We evaluate all models using perplexity (PP), as well as accuracy of the top prediction (Acc) and the top five predictions (Acc@5). The results are summarized in Table 2.

We can confirm that for code suggestion neural models outperform $n$-gram language models by a large margin. Furthermore, adding attention improves the results substantially ($2.3$ lower perplexity and $3.4$ percentage points increased accuracy). Interestingly, this increase can be attributed to a superior prediction of identifiers, which increased from an accuracy of $2.1\%$ to $21.4\%$. An LSTM with an attention window of $50$ gives us the best accuracy for the top prediction. We achieve further improvements for perplexity and accuracy of the top five predictions by using a sparse pointer network that uses a smaller memory of the past $20$ identifier representations.

### 5.1 QUALITATIVE ANALYSIS

Figures 3a-d show a code suggestion example involving an identifier usage. While the LSTM baseline is uncertain about the next token, we get a sensible prediction by using attention or the sparse pointer network. The sparse pointer network provides more reasonable alternative suggestions beyond the correct top suggestion.

Figures 3e-h show the use-case referring to a class attribute declared $67$ tokens in the past. Only the Sparse Pointer Network makes a good suggestion. Furthermore, the attention weights in 3i demonstrate that this model distinguished attributes from other groups of identifiers. We give a full example of a token-by-token suggestion of the Sparse Pointer Network in Figure 4 in the Appendix.

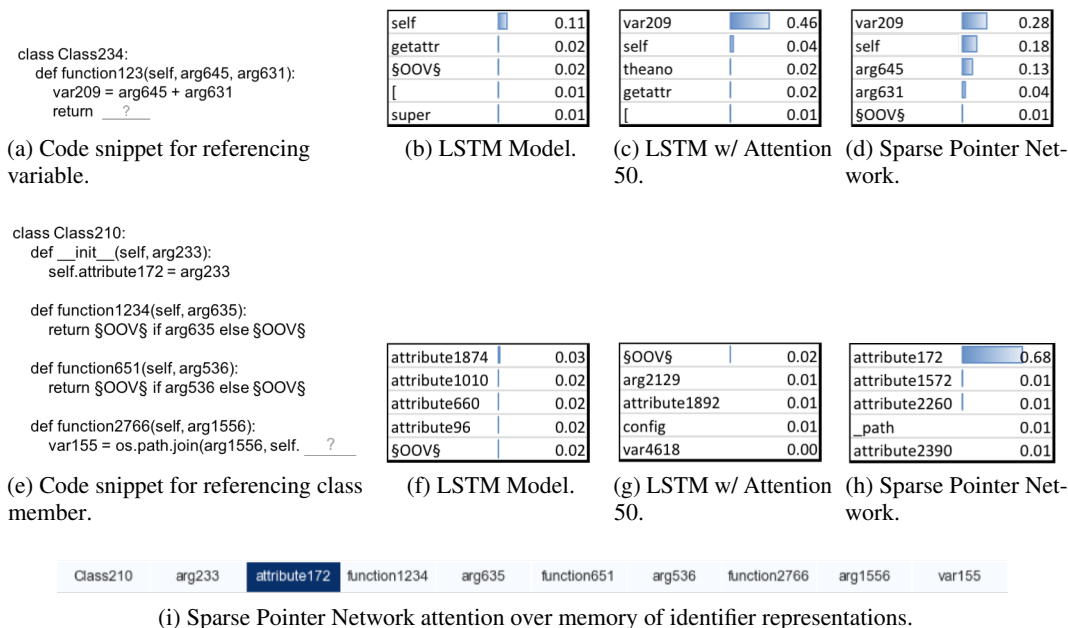

(a) Code snippet for referencing variable.

(b) LSTM Model.

(c) LSTM w/ Attention 50.

(d) Sparse Pointer Network.

(e) Code snippet for referencing class member.

(f) LSTM Model.

(g) LSTM w/ Attention 50.

(h) Sparse Pointer Network.

(i) Sparse Pointer Network attention over memory of identifier representations.

Figure 3: Code suggestion example involving a reference to a variable (a-d), a long-range dependency (e-h), and the attention weights of the Sparse Pointer Network (i).

# 6 RELATED WORK

Previous code suggestion work using methods from statistical NLP has mostly focused on $n$-gram models. Much of this work is inspired by Hindle et al. (2012) who argued that real programs fall into a much smaller space than the flexibility of programming languages allows. They were able to capture the repetitiveness and predictable statistical properties of real programs using language models. Subsequently, Tu et al. (2014) improved upon Hindle et al.'s work by adding a cache mechanism that allowed them to exploit locality stemming from the specialisation and decoupling of program modules. Tu et al.'s idea of adding a cache mechanism to the language model is specifically designed to exploit the properties of source code, and thus follows the same aim as the sparse attention mechanism introduced in this paper.

While the majority of preceding work trained on small corpora, Allamanis & Sutton (2013) created a corpus of 352M lines of Java code which they analysed with $n$-gram language models. The size of the corpus allowed them to train a single language model that was effective across multiple different project domains. White et al. (2015) later demonstrated that neural language models outperform $n$-gram models for code suggestion. They compared various $n$-gram models (up to nine grams), including Tu et al.'s cache model, with a basic RNN neural language model. Khanh Dam et al. (2016) compared White et al.'s basic RNN with LSTMs and found that the latter are better at code suggestion due to their improved ability to learn long-range dependencies found in source code. Our paper extends this line of work by introducing a sparse attention model that captures even longer dependencies.

The combination of lagged attention mechanisms with language modelling is inspired by Cheng et al. (2016) who equipped LSTM cells with a fixed-length memory tape rather than a single memory cell. They achieved promising results on the standard Penn Treebank benchmark corpus (Marcus et al., 1993). Similarly, Tran et al. added a *memory block* to LSTMs for language modelling of English, German and Italian and outperformed both $n$-gram and neural language models. Their memory encompasses representations of all possible words in the vocabulary rather than providing a sparse view as we do. Attention mechanisms were previously applied to the study of source code by Allamanis et al. who used a convolutional neural network combined with an attention mechanism to generate method names from bodies.

An alternative to our purely lexical approach to code suggestion involves the use of probabilistic context-free grammars (PCFGs) which exploit the formal grammar specifications and well-defined, deterministic parsers available for source code. These were used by Allamanis & Sutton (2014) to extract idiomatic patterns from source code. A weakness of PCFGs is their inability to model context-dependent rules of programming languages such as that variables need to be declared before being used. Maddison & Tarlow (2014) added context-aware variables to their PCFG model in order to capture such rules.

Ling et al. (2016) recently used a pointer network to generate code from natural language descriptions. Our use of a controller for deciding whether to generate from a language model or copy an identifier using a sparse pointer network is inspired by their latent code predictor. However, their inputs (textual descriptions) are short whereas code suggestion requires capturing very long-range dependencies that we addressed by a filtered view on the memory of previous identifier representations.

# 7 CONCLUSIONS AND FUTURE WORK

In this paper, we investigated neural language models for code suggestion of the dynamically-typed programming language Python. We released a corpus of 41M lines of Python crawled from GitHub and compared $n$-gram, standard neural language models, and attention. By using attention, we observed an order of magnitude more accurate prediction of identifiers. Furthermore, we proposed a sparse pointer network that can efficiently capture long-range dependencies by only operating on a filtered view of a memory of previous identifier representations. This model achieves the lowest perplexity and best accuracy among the top five predictions. The Python corpus and the code for our models is released at `https://github.com/uclmr/pycodesuggest`.

The presented methods were only tested for code suggestion within the same Python file. We are interested in scaling the approach to the level of entire code projects and collections thereof, as well as integrating a trained code suggestion model into an existing IDE. Furthermore, we plan to work on code completion, *i.e.*, models that provide a likely continuation of a partial token, using character language models (Graves, 2013).

ACKNOWLEDGMENTS

This work was supported by Microsoft Research through its PhD Scholarship Programme, an Allen Distinguished Investigator Award, and a Marie Curie Career Integration Award.

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

APPENDIX

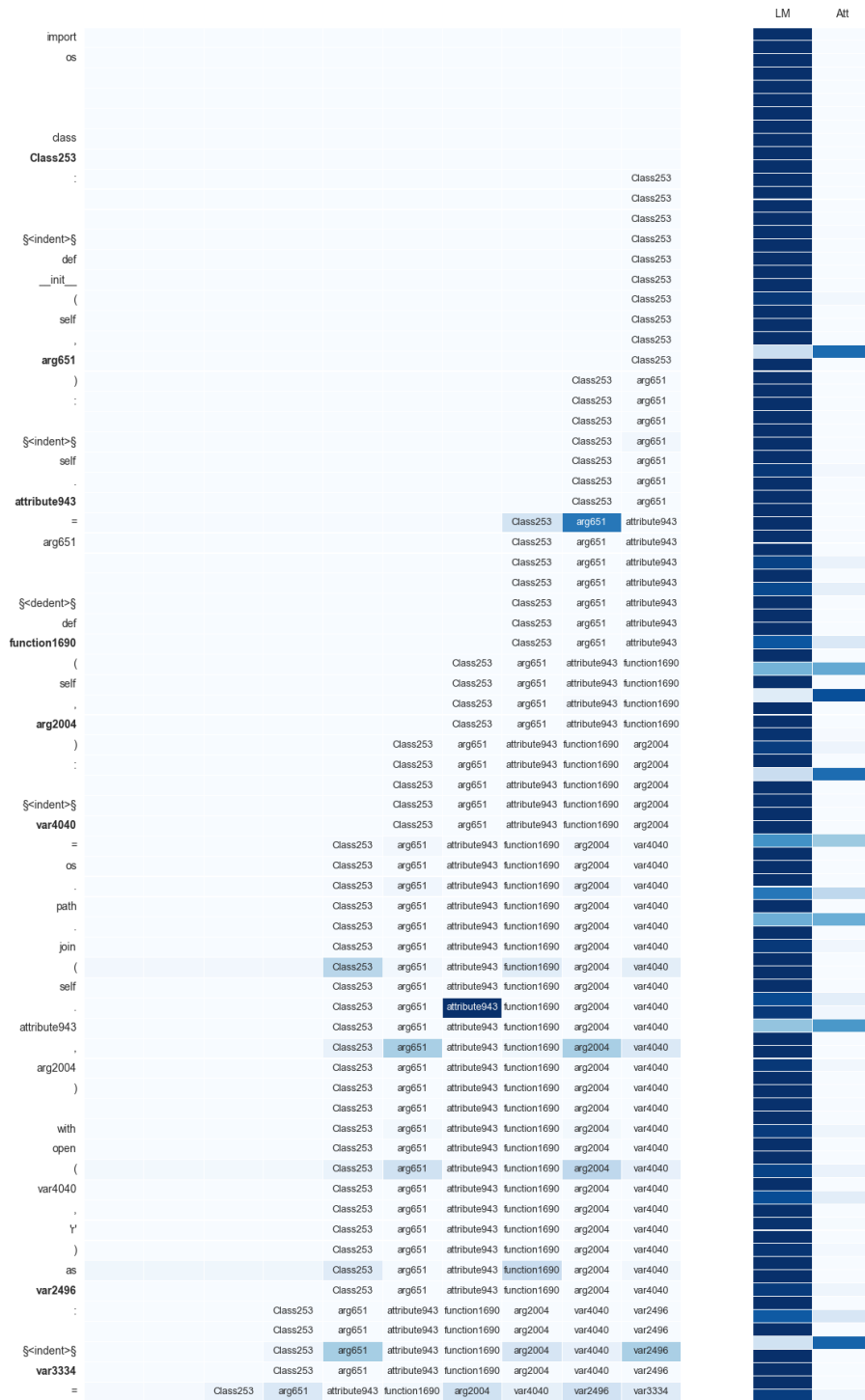

Figure 4: Full example of code suggestion with a Sparse Pointer Network. Boldface tokens on the left show the first declaration of an identifier. The middle part visualizes the memory of representations of these identifiers. The right part visualizes the output $\lambda$ of the controller, which is used for interpolating between the language model (LM) and the attention of the pointer network (Att).

