# Peer review of "Learning Python Code Suggestion with a Sparse Pointer Network"

_ICLR 2017 — rejected_

[Public Comment · (anonymous) · 07 Dec 2016]
**Prior Work & Suggestions**

Some of the claimed contributions the paper makes already exist in prior work. For instance:

→ There is already a Python data set available:

[Official Review · AnonReviewer1 · rating 5 · confidence 4 · 16 Dec 2016]
**An attention mechanism that isn't learned**

This paper takes a standard auto-regressive model of source code and augments it with a fixed attention policy that tracks the use of certain token types, like identifiers. Additionally they release a Python open source dataset. As expected this augmentation, the fixed attention policy, improves the perplexity of the model. It seems important to dig a bit deeper into these results and show the contribution of different token types to the achieve perplexity. This is alluded to in the text, but a more thorough comparison would be welcome. The idea of an attention policy that takes advantage of expert knowledge is a nice contribution, but perhaps if limited novelty --- for example the Maddison and Tarlow 2014 paper, which the authors cite, has scoping rules that track previously used identifiers in scope.

[Official Review · AnonReviewer2 · rating 6 · confidence 4 · 17 Dec 2016]

This paper uses a pointer network over a sparse window of identifiers to improve code suggestion for dynamically-typed languages. Code suggestion seems an area where attention and/or pointers truly show an advantage in capturing long term dependencies.

The sparse pointer method does seem to provide better results than attention for similar window sizes - specifically, comparing a window size of 20 for the attention and sparse pointer method shows the sparse pointer winning fairly definitively across the board. Given a major advantage of the pointer method is being able to use a large window size well thanks to the supervision the pointer provides, it was unfortunate (though understandable due to potential memory issues) not to see larger window sizes. Having a different batch size for the sparse pointer and attention models is unfortunate given it complicates an otherwise straight comparison between the two models.

The construction and filtering of the Python corpus sounds promising but as of now it is still inaccessible (listed in the paper as TODO). Given that code suggestion seems an interesting area for future long term dependency work, it may be promising as an avenue for future task exploration.

Overall this paper and the dataset are likely an interesting contribution even though there are a few potential issues.

[Official Review · AnonReviewer3 · rating 6 · confidence 4 · 18 Dec 2016]
**No Title**

This paper presents an improved neural language models designed for selected long-term dependency, i.e., to predict more accurately the next identifier for the dynamic programming language such as Python. The improvements are obtained by:

1) replacing the fixed-widow attention with a pointer network, in which the memory only consists of context representation of the previous K identifies introduced for the entire history. 
2) a conventional neural LSTM-based language model is combined with such a sparse pointer network with a controller, which linearly combines the prediction of both components using a dynamic weights, decided by the input, hidden state, and the context representations at the time stamp.

Such a model avoids the the need of large window size of the attention to predict next identifier, which usually requires a long-term dependency in the programming language. This is partly validated by the python codebase (which is another contribution of this paper) experiments in the paper.

While the paper still misses some critical information that I would like to see, including how the sparse pointer network performance chances with different size of K, and how computationally efficient it is for both training and inference time compared to LSTM w/ attention of various window size, and ablation experiments about how much (1) and (2) contribute respectively, it might be of interest to the ICLR community to see it accepted.

[Final Decision · Program Chairs · 06 Feb 2017]
**ICLR committee final decision**

This paper augments language models with attention to to capture long range dependencies through a sparse pointer network that is restricted to previously introduced identifiers, and demonstrates the proposed architecture over a new, released large-scale code suggestion corpus of 41M lines of Python code. The addition of long range attention over 20 identifiers improves perplexity compared to an LSTM with an attentional context of 50 words, but degrades accuracy (hit @1), while improving hit@5.
 The experimental validation however requires a more thorough analysis and more detailed ablation experiments and discussions, and more thorough comparison to related work. As is, many choices seem quite arbitrary and make it hard to determine if the model is really performing well (minibatch sizes, size of the memory for the LSTM, choice and number of identifiers for the sparse pointers, etc).